# Set2Graph: Learning Graphs From Sets

**Hadar Serviansky**[1]  **Nimrod Segol**[1]  **Jonathan Shlomi**[1]  **Kyle Cranmer**[2]

**Eilam Gross**[1]  **Haggai Maron**[3]  **Yaron Lipman**[1]

[1]Weizmann Institute of Science  [2]New York University  [3]NVIDIA Research

## Abstract

Many problems in machine learning can be cast as learning functions from sets to graphs, or more generally to hypergraphs; in short, Set2Graph functions. Examples include clustering, learning vertex and edge features on graphs, and learning features on triplets in a collection.

A natural approach for building Set2Graph models is to characterize all linear equivariant set-to-hypergraph layers and stack them with non-linear activations. This poses two challenges: (i) the expressive power of these networks is not well understood; and (ii) these models would suffer from high, often intractable computational and memory complexity, as their dimension grows exponentially.

This paper advocates a family of neural network models for learning Set2Graph functions that is both practical and of maximal expressive power (universal), that is, can approximate arbitrary continuous Set2Graph functions over compact sets. Testing these models on different machine learning tasks, mainly an application to particle physics, we find them favorable to existing baselines.

## 1 Introduction

We consider the problem of learning functions taking sets of vectors in $\mathbb{R}^{d_{\text{in}}}$ to graphs, or more generally hypergraphs; we name this problem Set2Graph, or set-to-graph. Set-to-graph functions appear in machine-learning applications such as clustering, predicting features on edges and nodes in graphs, and learning $k$-edge information in sets.

Mathematically, we represent each set-to-graph function as a collection of set-to-$k$-edge functions, where each set-to-$k$-edge function learns features on $k$-edges. That is, given an input set $\mathcal{X} = \{\boldsymbol{x}_1, \ldots, \boldsymbol{x}_n\} \subset \mathbb{R}^{d_{\text{in}}}$ we consider functions $\mathsf{F}^k$ at-

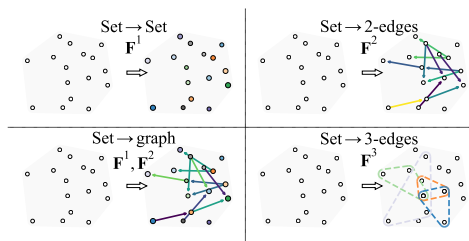

Figure 1: Set-to-graph functions are represented as collections of set-to-k-edge functions.

taching feature vectors to $k$-edges: each $k$-tuple $(\boldsymbol{x}_{i_1}, \ldots, \boldsymbol{x}_{i_k})$ is assigned with an output vector $\mathsf{F}^k(\mathcal{X})_{i_1, i_2, \ldots, i_k, :} \in \mathbb{R}^{d_{\text{out}}}$. Now, functions mapping sets to hypergraphs with hyper-edges of size up-to $k$ are modeled by $(\mathsf{F}^1, \mathsf{F}^2, \ldots, \mathsf{F}^k)$. For example, functions mapping sets to standard graphs are represented by $(\mathsf{F}^1, \mathsf{F}^2)$, see Figure 1.

Set-to-graph functions are well-defined if they satisfy a property called *equivariance* (defined later), and therefore the set-to-graph problem is an instance of the bigger class of equivariant learning [3, 27, 16]. A natural approach for learning equivariant set-to-graph model is using out-of-the-box full equivariant model as in [20].

A central question is: Are equivariant models *universal* for set-to-graph functions? That is, can equivariant models approximate any continuous equivariant function? In equivariant learning literature set-to-set models [44, 25] are proven equivariant universal [11, 30, 28]. In contrast, the situation for graph-to-graph equivariant models is more intricate: some models, such as message passing (a.k.a. graph convolutional networks), are known to be non-universal [41, 23, 19, 2], while high-order equivariant models are known to be universal [21] but require using high order tensors and therefore not practical. Universality of equivariant set-to-graph models is not known, as far as we are aware. In particular, are high order tensors required for universality (as the graph-to-graph case), or low order tensors (as in the set-to-set case) are sufficient?

In this paper we: (i) show that low order tensors are sufficient for set-to-graph universality, and (ii) build an equivariant model for the set-to-graph problem that is both *practical* (i.e., small number of parameters and no-need to build high-order tensors in memory) and *provably universal*. We achieve that with a composition of three networks: $\mathbf{F}^k = \boldsymbol{\psi} \circ \boldsymbol{\beta} \circ \boldsymbol{\phi}$, where $\boldsymbol{\phi}$ is a set-to-set model, $\boldsymbol{\beta}$ is a *non-learnable* broadcasting set-to-graph layer, and $\boldsymbol{\psi}$ is a simple graph-to-graph network using only a single Multi-Layer Perceptron (MLP) acting on each $k$-edge independently.

Our main motivation for this work comes from an important set-to-2-edges learning problem in particle physics: partitioning (clustering) of simulated particles generated in the Large Hadron Collider (LHC). We demonstrate our model produces state of the art results on this task compared to relevant baselines. We also experimented with another set-to-2-edges problem of Delaunay triangulation, and a set-to-3-edges problem of 3D convex hull, in which we also achieve superior performances to the baselines.

## 2 Previous work

**Equivariant learning.** In many learning setups the task is invariant or equivariant to certain transformations of the input. The Canonical example is image recognition tasks [18, 17] and set classification tasks [44, 25]. Earlier methods such as [34] used non-equivariant methods to learn set functions. Restricting models to be invariant or equivariant to these transformation was shown to be an excellent approach for reducing the number of parameters of models while improving generalization [44, 25, 13, 9, 33, 41, 15, 20, 19, 3, 5, 7, 40, 4, 8, 37, 39, 37]. There has been a keen interest in the analysis of equivariant models [27, 15], especially the analysis of their approximation power [44, 25, 21, 11, 30, 22]. As far we know, the set-to-graph case was not treated before.

**Similarity learning.** Our work is related to the field of similarity learning, in which the goal is to learn a similarity function on pairs of inputs. In most cases, a siamese architecture is used in order to extract features for each input and then a similarity score is calculated based on this pair of features [1, 31, 43]. The difference from our setup is that similarity learning is aimed at extracting pairwise relations between two inputs, independently from the other members of the set, while we learn these pairwise relations globally from the entire input set. In the experimental section we show that the independence assumption taken in similarity learning might cause a significant degradation in performance compared to our global approach.

**Other related methods.** [10] suggest a method for meta-clustering that can be seen as an instance of the set2graph setup. Their method is based on LSTMs and therefore depends on the order of the set elements. In contrast, our method is blind (equivariant) to the chosen order of the input sets. The Neural relational Inference model [12] is another related work that targets learning relations and dynamics of a set of objects in an unsupervised manner. [35] had previously tackled planar Delaunay triangulation and convex hull prediction problems using a non-equivariant network.

## 3 Learning hypergraphs from sets

We would like to learn functions of sets of $n$ vectors in $\mathbb{R}^{d_{\text{in}}}$ to hypergraphs with $n$ nodes (think of the nodes as corresponding to the set elements), and arbitrary $k$-edge feature vectors in $\mathbb{R}^{d_{\text{out}}}$, where a $k$-edge is defined as a $k$-tuple of set elements. A function mapping sets of vectors to $k$-edges is called set-to-$k$-edge function and denoted $\mathbf{F}^k : \mathbb{R}^{n \times d_{\text{in}}} \to \mathbb{R}^{n^k \times d_{\text{out}}}$. Consequently, a set-to-hypergraph function would be modeled as a sequence $(\mathbf{F}^1, \mathbf{F}^2, \ldots, \mathbf{F}^K)$, for target hypergraphs with hyperedges of maximal size $K$. For example, $\mathbf{F}^2$ learns pairwise relations in a set; and $(\mathbf{F}^1, \mathbf{F}^2)$ is a function from sets to graphs (outputs both node features and pairwise relations); see Figure 1.

**Our goal** is to design permutation equivariant neural network models for $\mathbf{F}^k$ that are as-efficient-as-possible in terms of number of parameters and memory usage, but on the same time with maximal expressive power, i.e., universal.

**Representing sets and $k$-edges.** A matrix $\boldsymbol{X} = (\boldsymbol{x}_1, \boldsymbol{x}_2, \ldots, \boldsymbol{x}_n)^T \in \mathbb{R}^{n \times d_{\mathrm{in}}}$ represents a set of $n$ vectors $\boldsymbol{x}_i \in \mathbb{R}^{d_{\mathrm{in}}}$ and therefore should be considered up to re-ordering of its rows. We denote by $S_n = \{\sigma\}$ the symmetric group, that is the group of bijections (permutations) $\sigma : [n] \to [n]$, where $[n] = \{1, \ldots, n\}$. We denote by $\sigma \cdot \boldsymbol{X}$ the matrix resulting in reordering the rows of $\boldsymbol{X}$ by the permutation $\sigma$, i.e., $(\sigma \cdot \boldsymbol{X})_{i,j} = \boldsymbol{X}_{\sigma^{-1}(i),j}$. In this notation, $\boldsymbol{X}$ and $\sigma \cdot \boldsymbol{X}$ represent the same set, for all permutations $\sigma$.

$k$-edges are represented as a tensor $\mathbf{Y} \in \mathbb{R}^{n^k \times d_{\mathrm{out}}}$, where $\mathbf{Y}_{\boldsymbol{i},:} \in \mathbb{R}^{d_{\mathrm{out}}}$ denotes the feature vector attached to the $k$-edge defined by the $k$-tuple $(\boldsymbol{x}_{i_1}, \boldsymbol{x}_{i_2}, \ldots, \boldsymbol{x}_{i_k})$, where $\boldsymbol{i} = (i_1, i_2, \ldots, i_k) \in [n]^k$ is a multi-index with non-repeating indices. Similarly to the set case, $k$-edges are considered up-to renumbering of the nodes by some permutation $\sigma \in S_n$. That is, if we define the action $\sigma \cdot \mathbf{Y}$ by $(\sigma \cdot \mathbf{Y})_{\boldsymbol{i},j} = \mathbf{Y}_{\sigma^{-1}(\boldsymbol{i}),j}$, where $\sigma^{-1}(\boldsymbol{i}) = (\sigma^{-1}(i_1), \sigma^{-1}(i_2), \ldots, \sigma^{-1}(i_k))$, then $\mathbf{Y}$ and $\sigma \cdot \mathbf{Y}$ represent the same $k$-edge data, for all $\sigma \in S_n$.

**Equivariance.** A sufficient condition for $\mathbf{F}^k$ to represent a well-defined map between sets $\boldsymbol{X} \in \mathbb{R}^{n \times d_{\mathrm{in}}}$ and $k$-edge data $\mathbf{Y} \in \mathbb{R}^{n^k \times d_{\mathrm{out}}}$ is *equivariance* to permutations, namely

$$\mathbf{F}^k(\sigma \cdot \boldsymbol{X}) = \sigma \cdot \mathbf{F}^k(\boldsymbol{X}), \tag{1}$$

for all sets $\boldsymbol{X} \in \mathbb{R}^{n \times d_{\mathrm{in}}}$ and permutations $\sigma \in S_n$. Equivariance guarantees, in particular, that the two equivalent sets $\boldsymbol{X}$ and $\sigma \cdot \boldsymbol{X}$ are mapped to equivalent $k$-edge data tensors $\mathbf{F}^k(\boldsymbol{X})$ and $\sigma \cdot \mathbf{F}^k(\boldsymbol{X})$.

**Set-to-$k$-edge models.** In this paper we explore the following equivariant neural network model family for approximating $\mathbf{F}^k$:

$$\mathbf{F}^k(\boldsymbol{X}; \theta) = \boldsymbol{\psi} \circ \boldsymbol{\beta} \circ \boldsymbol{\phi}(\boldsymbol{X}), \tag{2}$$

where $\boldsymbol{\phi}, \boldsymbol{\beta}$, and $\boldsymbol{\psi}$ will be defined soon. For $\mathbf{F}^k$ to be equivariant (as in equation 1) it is sufficient that its constituents, namely $\boldsymbol{\phi}, \boldsymbol{\beta}, \boldsymbol{\psi}$, are equivariant. That is, $\boldsymbol{\phi}, \boldsymbol{\beta}, \boldsymbol{\psi}$ all satisfy equation 1.

**Set-to-graphs models.** Given the model of set-to-$k$-edge functions, a model for a set-to-graph function can now be constructed from a pair of set-to-$k$-edge networks $(\mathbf{F}^1, \mathbf{F}^2)$. Similarly, set-to-hypergraph function would require $(\mathbf{F}^1, \ldots, \mathbf{F}^K)$, where $K$ is the maximal hyperedge size. Figure 1 shows an illustration of set-to-$k$-edge and set-to-graph functions

**$\boldsymbol{\phi}$ component.** $\boldsymbol{\phi} : \mathbb{R}^{n \times d_{\mathrm{in}}} \to \mathbb{R}^{n \times d_1}$ is a set-to-set equivariant model, that is $\boldsymbol{\phi}$ is mapping sets of vectors in $\mathbb{R}^{d_{\mathrm{in}}}$ to sets of vectors in $\mathbb{R}^{d_1}$. To achieve the universality goal we will need $\boldsymbol{\phi}$ to be universal as set-to-set model; that is, $\boldsymbol{\phi}$ can approximate arbitrary continuous set-to-set functions. Several options exists [11, 28] although probably the simplest option is either DeepSets [44] or one of its variations; all were proven to be universal recently in [30].

In practice, as will be clear later from the proof of the universality of the model, when building set-to-graph or set-to-hypergraph model, the $\boldsymbol{\phi}$ (set-to-set) part of the $k$-edge networks can be shared between different set-to-$k$-edge models, $\mathbf{F}^k$, without compromising universality.

**$\boldsymbol{\beta}$ component.** $\boldsymbol{\beta} : \mathbb{R}^{n \times d_1} \to \mathbb{R}^{n^k \times d_2}$ is a non-learnable linear *broadcasting layer* mapping sets to $k$-edges. In theory, as shown in [20] the space of equivariant linear mappings $\mathbb{R}^{n \times d_1} \to \mathbb{R}^{n^k \times d_2}$ is of dimension $d_1 d_2 \mathrm{bell}(k+1)$ which can be very high since $\mathrm{bell}$ numbers have exponential growth. Interestingly, in the set-to-$k$-edge case one can achieve universality with only $k$ linear operators. We define the broadcasting operator to be

$$\boldsymbol{\beta}(\boldsymbol{X})_{\boldsymbol{i},:} = [\boldsymbol{x}_{i_1}, \boldsymbol{x}_{i_2}, \ldots, \boldsymbol{x}_{i_k}], \tag{3}$$

where $\boldsymbol{i} = (i_1, \ldots, i_k)$ and brackets denote concatenation in the feature dimension, that is, for $\boldsymbol{A} \in \mathbb{R}^{n^k \times d_a}$, $\boldsymbol{B} \in \mathbb{R}^{n^k \times d_b}$ their concatenation is $[\boldsymbol{A}, \boldsymbol{B}] \in \mathbb{R}^{n^k \times (d_a + d_b)}$. Therefore, the feature output dimension of $\boldsymbol{\beta}$ is $d_2 = k d_1$.

As an example, consider the graph case, where $k = 2$. In this case $\boldsymbol{\beta}(\boldsymbol{X})_{i_1,i_2,:} = [\boldsymbol{x}_{i_1}, \boldsymbol{x}_{i_2}]$. This function is illustrated in Figure 2 broadcasting data in $\mathbb{R}^{n \times d_1}$ to tensor $\mathbb{R}^{n \times n \times d_2}$.

To see that the broadcasting layer is equivariant, it is enough to consider a single feature $\boldsymbol{\beta}(\boldsymbol{X})_{\boldsymbol{i}} = \boldsymbol{x}_{i_1}$. Permuting the rows of $\boldsymbol{X}$ by a permutation $\sigma$ we get $\boldsymbol{\beta}(\sigma \cdot \boldsymbol{X})_{\boldsymbol{i},j} = \boldsymbol{x}_{\sigma^{-1}(i_1),j} = \boldsymbol{\beta}(\boldsymbol{X})_{\sigma^{-1}(\boldsymbol{i}),j} = (\sigma \cdot \boldsymbol{\beta}(\boldsymbol{X}))_{\boldsymbol{i},j}$.

**$\psi$ component.** $\quad \psi : \mathbb{R}^{n^k \times d_2} \to \mathbb{R}^{n^k \times d_{\text{out}}}$ is a mapping of $k$-tensors to $k$-tensors. Here the theory of equivariant operators indicates that the space of linear equivariant maps is of dimension $d_2 d_{\text{out}} \text{bell}(2k)$ that suggests a huge number of model parameters even for a single linear layer. Surprisingly, universality can be achieved with much less, in fact a single linear operator (i.e., scaled identity) in each layer. In the multifeature multi-layer case this boils to applying a

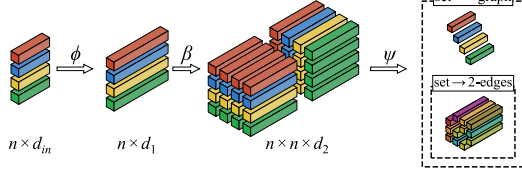

Figure 2: The model architecture for the Set-to-graph and set-to-2-edge functions.

Multi-Layer Perceptron $\boldsymbol{m} : \mathbb{R}^{d_2} \to \mathbb{R}^{d_{\text{out}}}$ independently to each feature vector in the input tensor $\mathsf{X} \in \mathbb{R}^{n^k \times d_2}$. That is, we use

$$\psi(\mathsf{X})_{\boldsymbol{i},:} = \boldsymbol{m}(\mathsf{X}_{\boldsymbol{i},:}). \tag{4}$$

Figure 2 illustrates set-to-2-edges and set-to-graph models incorporating the three components $\phi, \boldsymbol{\beta}, \psi$ discussed above. We note that, Indeed, $\phi, \boldsymbol{\beta}, \psi$ are equivariant.

## 4 Universality of set-to-graph models

In this section we prove that the model $\mathsf{F}^k$ introduced above, is universal, in the sense it can approximate arbitrary continuous equivariant set-to-$k$-edge functions $\mathsf{G}^k : \mathbb{R}^{n \times d_{\text{in}}} \to \mathbb{R}^{n^k \times d_{\text{out}}}$ over compact domains $K \subset \mathbb{R}^{n \times d_{\text{in}}}$.

**Theorem 1.** *The model $\mathsf{F}^k$ is set-to-$k$-edge universal.*

A corollary of Theorem 1 establishes set-to-hypergraph universal models:

**Theorem 2.** *The model $(\mathsf{F}^1, \dots, \mathsf{F}^k)$ is set-to-hypergraph universal.*

Our main tool for proving Theorem 1 is a characterization of the equivariant set-to-$k$-edge *polynomials* $\mathsf{P}^k$. This characterization can be seen as a generalization of the characterization of set-to-set equivariant polynomial recently appeared in [30].

We consider an arbitrary set-to-$k$-edge continuous mapping $\mathsf{G}^k(\boldsymbol{X})$ over a compact set $K \subset \mathbb{R}^{n \times d_{\text{in}}}$. Since $\mathsf{G}^k$ is equivariant we can assume $K$ is symmetric, i.e., $\sigma \cdot K = K$ for all $\sigma \in S_n$. The proof consists of three parts: (i) Characterization of the equivariant set-to-$k$-edge polynomials $\mathsf{P}^k$. (ii) Showing that every equivariant continuous set-to-$k$-edge function $\mathsf{G}^k$ can be approximated by some $\mathsf{P}^k$. (iii) Every $\mathsf{P}^k$ can be approximated by our model $\mathsf{F}^k$.

Before providing the full proof which contains some technical derivations let us provide a simpler universality proof (under some mild extra conditions) for the set-to-2-edge model, $\mathsf{F}^2$, based on the Singular Value Decomposition (SVD).

### 4.1 A simple proof for universality of second-order tensors

It is enough to consider the $d_{\text{out}} = 1$ case; the general case is implied by applying the argument for each output feature dimension independently. Let $\mathsf{G}^2$ be an arbitrary continuous equivariant set-to-2-edge function $\mathsf{G}^2 : K \subset \mathbb{R}^{n \times d_{\text{in}}} \to \mathbb{R}^{n \times n}$. We want to approximate $\mathsf{G}^2$ with our model $\mathsf{F}^2$. First, note that without losing generality we can assume $\mathsf{G}^2(\boldsymbol{X})$ has a simple spectrum (i.e., eigenvalues are all different) for all $\boldsymbol{X} \in K$. Indeed, if this is not the case we can always choose $\lambda > 0$ sufficiently large and consider $\mathsf{G}^2 + \lambda \text{diag}(1, 2, \dots, n)$. This diagonal addition does not change

the 2-edge values assigned by $\mathbf{G}^2$, and it guarantees a simple spectrum using standard hermitian matrix eigenvalue perturbation theory (see e.g., [32], Section IV:4).

Now let $\mathbf{G}^2(\boldsymbol{X}) = \boldsymbol{U}(\boldsymbol{X})\boldsymbol{\Sigma}(\boldsymbol{X})\boldsymbol{V}(\boldsymbol{X})^T$ be the SVD of $\mathbf{G}^2(\boldsymbol{X})$, where $\boldsymbol{U} = [\boldsymbol{u}_1, \ldots, \boldsymbol{u}_n]$ and $\boldsymbol{V} = [\boldsymbol{v}_1, \ldots, \boldsymbol{v}_n]$. Since $\mathbf{G}^2(\boldsymbol{X})$ has a simple spectrum, $\boldsymbol{U}, \boldsymbol{V}, \boldsymbol{\Sigma}$ are all continuous in $\boldsymbol{X}$; $\boldsymbol{\Sigma}$ is unique, and $\boldsymbol{U}, \boldsymbol{V}$ are unique up to a sign flip of the singular vectors (i.e., columns of $\boldsymbol{U}, \boldsymbol{V}$) [24]. Let us first assume that the singular vectors can be chosen uniquely also up to a sign, later we show how we achieve this with some additional mild assumption.

Now, uniqueness of the SVD together with the equivariance of $\mathbf{G}^2$ imply that $\boldsymbol{U}, \boldsymbol{V}$ are continuous *set-to-set* equivariant and $\boldsymbol{\Sigma}$ is a continuous *set invariant* function:

$$
\begin{aligned}
(\sigma \cdot \boldsymbol{U}(\boldsymbol{X}))\boldsymbol{\Sigma}(\boldsymbol{X})(\sigma \cdot \boldsymbol{V}(\boldsymbol{X}))^T \\
= \sigma \cdot \boldsymbol{G}(\boldsymbol{X}) = \boldsymbol{G}(\sigma \cdot \boldsymbol{X}) \\
= \boldsymbol{U}(\sigma \cdot \boldsymbol{X})\boldsymbol{\Sigma}(\sigma \cdot \boldsymbol{X})\boldsymbol{V}(\sigma \cdot \boldsymbol{X})^T.
\end{aligned}
\tag{5}
$$

Lastly, since $\phi$ is set-to-set universal there is a choice of its parameters so that it approximates arbitrarily well the equivariant set-to-set function $\boldsymbol{Y} = [\boldsymbol{U}, \boldsymbol{V}, \mathbf{1}\mathbf{1}^T\boldsymbol{\Sigma}]$. The $\psi$ component can be chosen by noting that $\mathbf{G}^2(\boldsymbol{X})_{i_1, i_2} = \sum_{j=1}^n \sigma_j \boldsymbol{U}_{i_1, j}\boldsymbol{V}_{i_2, j} = p(\beta(\boldsymbol{Y})_{i_1, i_2, :})$, where $\sigma_j$ are the singular values, and $p : \mathbb{R}^{6n} \to \mathbb{R}$ is a cubic polynomial. To conclude pick $\boldsymbol{m}$ to approximate $\boldsymbol{p}$ sufficiently well so that $\psi \circ \beta \circ \phi$ approximates $\mathbf{G}^2$ to the desired accuracy.

To achieve uniqueness of the singular vectors up-to a sign we can add, e.g., the following assumption: $\mathbf{1}^T\boldsymbol{u}_i(\boldsymbol{X}) \neq 0 \neq \mathbf{1}^T\boldsymbol{v}_i(\boldsymbol{X})$ for all singular vectors and $\boldsymbol{X} \in K$. Using this assumption we can always pick $\boldsymbol{u}_i(\boldsymbol{X}), \boldsymbol{v}_i(\boldsymbol{X})$ in the SVD so that $\mathbf{1}^T\boldsymbol{u}_i(\boldsymbol{X}) > 0, \mathbf{1}^T\boldsymbol{v}_i(\boldsymbol{X}) > 0$, for all $i \in [n]$. Lastly, note that equation 5 suggests that also outer-product can be used as a broadcasting layer. We now move to the general proof.

### 4.2 Equivariant set-to-$k$-edge polynomials

We start with a characterization of the set-to-$k$-edge equivariant polynomials $\mathbf{P}^k : \mathbb{R}^{n \times d_{\text{in}}} \to \mathbb{R}^{n^k \times d_{\text{out}}}$. We need some more notation. Given a vector $\boldsymbol{x} \in \mathbb{R}^d$, and a multi-index $\alpha \in [n]^d$, we set $\boldsymbol{x}^\alpha = \prod_{i=1}^d x_i^{\alpha_i}$; $|\alpha| = \sum_{i=1}^d \alpha_i$; and define accordingly $\boldsymbol{X}^\alpha = (\boldsymbol{x}_1^\alpha, \ldots, \boldsymbol{x}_n^\alpha)^T$. Given two tensors $\mathbf{A} \in \mathbb{R}^{n^{k_1}}$, $\mathbf{B} \in \mathbb{R}^{n^{k_2}}$ we use the notation $\mathbf{A} \otimes \mathbf{B} \in \mathbb{R}^{n^{k_1+k_2}}$ to denote the tensor-product, defined by $(\mathbf{A} \otimes \mathbf{B})_{\boldsymbol{i}_1, \boldsymbol{i}_2} = \mathbf{A}_{\boldsymbol{i}_1}\mathbf{B}_{\boldsymbol{i}_2}$, where $\boldsymbol{i}_1, \boldsymbol{i}_2$ are suitable multi-indices. Lastly, we denote by $\boldsymbol{\alpha} = (\alpha^1, \ldots, \alpha^k)$ a vector of multi-indices $\alpha^i \in [n]^d$, and $\boldsymbol{X}^{\boldsymbol{\alpha}} = \boldsymbol{X}^{\alpha^1} \otimes \cdots \otimes \boldsymbol{X}^{\alpha^k}$.

**Theorem 3.** *An equivariant set-to-$k$-edge polynomial* $\mathbf{P}^k : \mathbb{R}^{n \times d_{in}} \to \mathbb{R}^{n^k \times d_{out}}$ *can be written as*

$$
\mathbf{P}^k(\boldsymbol{X}) = \sum_{\boldsymbol{\alpha}} \boldsymbol{X}^{\boldsymbol{\alpha}} \otimes \boldsymbol{q}_{\boldsymbol{\alpha}}(\boldsymbol{X})
\tag{6}
$$

*where* $\boldsymbol{\alpha} = (\alpha^1, \ldots, \alpha^k)$, $\alpha^i \in [n]^{d_{in}}$, *and* $\boldsymbol{q}_{\boldsymbol{\alpha}} : \mathbb{R}^{n \times d_{in}} \to d_{out}$ *are* $S_n$ *invariant polynomials.*

As an example, consider the graph case, where $k = 2$. Equivariant set-to-2-edge polynomials take the form:

$$
\mathbf{P}^k(\boldsymbol{X}) = \sum_{\alpha^1, \alpha^2} \boldsymbol{X}^{\alpha^1} \otimes \boldsymbol{X}^{\alpha^2} \otimes \boldsymbol{q}_{\alpha^1, \alpha^2}(\boldsymbol{X}),
\tag{7}
$$

and coordinate-wise

$$
\mathbf{P}^k_{ijl}(\boldsymbol{X}) = \sum_{\alpha^1, \alpha^2} \boldsymbol{x}_i^{\alpha_1} \boldsymbol{x}_j^{\alpha_2} q_{\alpha_1, \alpha_2, l}(\boldsymbol{X}).
\tag{8}
$$

The general proof idea and proof itself is given in the supplementary. Figure 3 provides an illustration of these polynomials.

**Approximating $\mathbf{G}^k$ with a polynomial $\mathbf{P}^k$.** We denote for an arbitrary tensor $\mathbf{A} \in \mathbb{R}^{a \times b \times \cdots \times c}$ its infinity norm by $\|\mathbf{A}\|_\infty = \max_{\boldsymbol{i}} |\mathbf{A}_{\boldsymbol{i}}|$.

**Lemma 1.** *Let* $\mathbf{G}^k : K \subset \mathbb{R}^{n \times d_{in}} \to \mathbb{R}^{n^k \times d_{out}}$ *be a continuous equivariant function over a symmetric domain* $K \subset \mathbb{R}^{n \times d_{out}}$. *For an arbitrary* $\epsilon > 0$, *there exists an equivariant polynomial* $\mathbf{P}^k : \mathbb{R}^{n \times d_{in}} \to \mathbb{R}^{n^k \times d_{out}}$ *so that*

$$
\max_{\boldsymbol{X} \in K} \left\| \mathbf{G}^k(\boldsymbol{X}) - \mathbf{P}^k(\boldsymbol{X}) \right\|_\infty < \epsilon.
$$

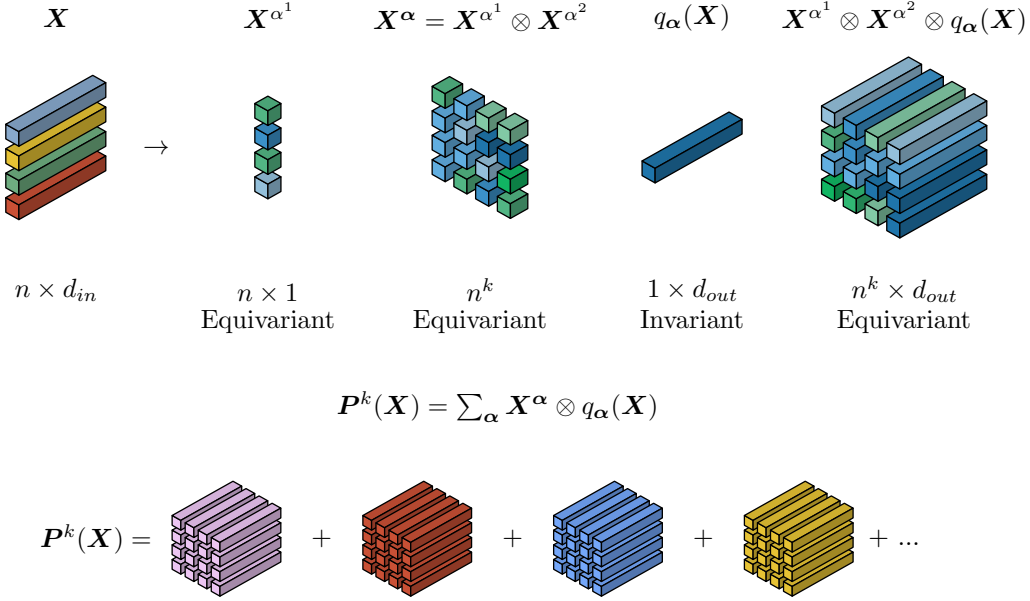

$$\boldsymbol{P}^k(\boldsymbol{X}) = \sum_{\boldsymbol{\alpha}} \boldsymbol{X}^{\boldsymbol{\alpha}} \otimes q_{\boldsymbol{\alpha}}(\boldsymbol{X})$$

Figure 3: Illustration of the structure of an equivariant set-to-$k$-edge polynomial $\mathbf{P}^k$, for $k = 2$.

This is a standard lemma, similar to [42, 21, 30]; we provide a proof in the supplementary.

**Approximating $\mathbf{P}^k$ with a network $\mathbf{F}^k$.** The final component of the proof of Theorem 1 is showing that an equivariant polynomial $\mathbf{P}^k$ can be approximated over $K$ using a network of the form in equation 2. The key is to use the characterization of Theorem 3 and write $\mathbf{P}^k$ in a similar form to our model in equation 2:

$$\mathbf{P}^k_{\boldsymbol{i},:}(\boldsymbol{X}) = p(\boldsymbol{\beta}(\boldsymbol{H}(\boldsymbol{X}))_{\boldsymbol{i},:}), \tag{9}$$

where $\boldsymbol{H} : K \to \mathbb{R}^{n \times d_1}$ defined by $\boldsymbol{H}(\boldsymbol{X})_{i,:} = [\boldsymbol{x}_i, \boldsymbol{q}(\boldsymbol{X})]$, where $\boldsymbol{q}(\boldsymbol{X}) = [\boldsymbol{q}_{\boldsymbol{\alpha}_1}(\boldsymbol{X}), \dots, \boldsymbol{q}_{\boldsymbol{\alpha}_m}(\boldsymbol{X})]$, and $\boldsymbol{\alpha}_1, \boldsymbol{\alpha}_2, \dots, \boldsymbol{\alpha}_m$ are all the multi-indices participating in the sum in equation 6. Note that

$$\boldsymbol{\beta}(\boldsymbol{H}(\boldsymbol{X}))_{\boldsymbol{i},:} = \left[\boldsymbol{x}_{i_1}, \boldsymbol{q}(\boldsymbol{X}), \boldsymbol{x}_{i_2}, \boldsymbol{q}(\boldsymbol{X}), \dots, \boldsymbol{x}_{i_k}, \boldsymbol{q}(\boldsymbol{X})\right].$$

Therefore, $\boldsymbol{p} : \mathbb{R}^{d_2} \to \mathbb{R}^{d_{\text{out}}}$ is chosen as the polynomial

$$\boldsymbol{p} : [\boldsymbol{x}_1, \boldsymbol{y}, \boldsymbol{x}_2, \boldsymbol{y}, \dots, \boldsymbol{x}_k, \boldsymbol{y}] \mapsto \sum_{\boldsymbol{\alpha}} \boldsymbol{x}_1^{\boldsymbol{\alpha}^1} \cdots \boldsymbol{x}_k^{\boldsymbol{\alpha}^k} \boldsymbol{y}_{\boldsymbol{\alpha}},$$

where $\boldsymbol{y} = [\boldsymbol{y}_{\boldsymbol{\alpha}_1}, \dots, \boldsymbol{y}_{\boldsymbol{\alpha}_m}] \in \mathbb{R}^{m d_{\text{out}}}$, and $\boldsymbol{y}_{\boldsymbol{\alpha}_i} \in \mathbb{R}^{d_{\text{out}}}$.

In view of equation 9 all we have left is to choose $\phi$ and $\psi$ (i.e., $\boldsymbol{m}$) to approximate $\boldsymbol{H}, \boldsymbol{p}$ (resp.) to a desired accuracy. We detail the rest of the proof in the supplementary.

**Universality of the set-to-hypergraph model.** Theorem 2 follows from Theorem 1 by considering a set-to-hypergraph continuous function $\mathbf{G}$ as a collection $\mathbf{G}^k$ of set-to-$k$-edge functions and approximating each one using our model $\mathbf{F}^k$. Note that universality still holds if $\mathbf{F}^1, \dots, \mathbf{F}^K$ all share the $\phi$ part of the network (assuming sufficient width $d_1$).

Note that a set-to-$k$-edge model (in equation 2) is not universal when approximating set-to-hypergraph functions:

**Proposition 1.** *The set-to-2-edge model, $\mathbf{F}^2$, cannot approximate general set-to-graph functions.*

The proof is in the supplementary; it shows that even the constant function that outputs 1 for 1-edges (nodes), and 0 for 2-edges cannot be approximated by a set-to-2-edge model $\mathbf{F}^2$.

# 5 Applications

## 5.1 Model variants and baselines

We tested our model on three learning tasks from two categories: set-to-2-edge and set-to-3-edge.

**Variants of our model.** We consider two variations of our model:

- **S2G**: This is Our basic model. We used the $\mathsf{F}^2$ and $\mathsf{F}^3$ (resp.) models for these learning tasks. for $\mathsf{F}^2$, $\phi$ is implemented using DeepSets [44] with 5 layers and output dimension $d_1 \in \{5, 80\}$; $\psi$ is implemented with an MLP, $m$, with $\{2, 3\}$ layers with input dimension $d_2$ defined by $d_1$ and $\beta$. $\beta$ is implemented according to equation 3: for $k = 2$ it uses $d_2 = 2 * d_1$ output features. For $\mathsf{F}^3$, S2G is described in section 5.4.

- **S2G+**: For the $k = 2$ case we have also tested a more general (but not more expressive) broadcasting $\beta$ defined using the full equivariant basis $\mathbb{R}^n \to \mathbb{R}^{n^2}$ from [20] that contains $\mathrm{bell}(3) = 5$ basis operations. This broadcasting layer gives $d_2 = 5 * d_1$.

**Baselines.** We compare our results to the following baselines:

- **MLP**: A standard multilayer perceptron applied to the flattened set features.

- **SIAM**: A popular similarity learning model (see e.g., [43]) based on Siamese networks. This model has the same structure as in equation 2 where $\phi$ is a Siamese MLP (a non-universal set-to-set function) that is applied independently to each element in the set. We use the same loss we use with our model (according to the task at hand).

- **SIAM-3**: The same architecture as **SIAM** but with a triplet loss [38] on the learned representations based on $l2$ distance, see e.g., [29]. Edge predictions are obtained by thresholding distances of pairs of learned representations.

- **GNN**: A Graph Neural Network [23] applied to the $k$-NN ($k \in \{0, 5, 10\}$) induced graph. Edge prediction is done via outer-product [14].

- **AVR**: A non-learnable geometric-based baseline called Adaptive Vertex Reconstruction [36] typically used for the particle physics problem we tackle. More information can be found in the supplementary material.

More architecture, implementation, hyper-parameter details and number of parameters can be found in the supplementary material.

## 5.2 Partitioning for particle physics

The first learning setup we tackle is learning set-to-2-edge functions. Here, each training example is a pair $(\boldsymbol{X}, \boldsymbol{Y})$ where $\boldsymbol{X}$ is a set $\boldsymbol{X} = (\boldsymbol{x}_1, \boldsymbol{x}_2, \ldots, \boldsymbol{x}_n)^T \in \mathbb{R}^{n \times d_{\mathrm{in}}}$ and $\boldsymbol{Y} \in \{0, 1\}^{n \times n}$ is an adjacency matrix (the diagonal of $\boldsymbol{Y}$ is ignored). Our main experiment tackles an important particle partitioning problem.

**Problem statement.** In particle physics experiments, such as the Large Hadron Collider (LHC), beams of incoming particles are collided at high energies. The results of the collision are outgoing particles, whose properties (such as the trajectory) are measured by detectors surrounding the collision point. A critical low-level task for analyzing this data is to associate the particle trajectories to their progenitor, which can be formalized as partitioning sets of particle trajectories into subsets according to their unobserved point of origin in space. This task is referred to as vertex reconstruction in particle physics and is illustrated in Figure 4. We cast this problem as a set-to-2-edge problem by treating the measured particle trajectories as elements in the input set and nodes in the output graph, where the parameters that characterize them serve as the node features. An edge between two nodes indicates that the two particles come from a common progenitor or vertex.

**Data.** We consider three different types (or *flavors*) of particle sets (called *jets*) corresponding to three different fundamental data generating processes labeled bottom-jets, charm-jets, and light-jets (B/C/L). The important distinction between the flavors is the typical number of partitions in each set. Since it is impossible to label real data collected in the detectors at the LHC, algorithms for particle physics are typically designed with high-fidelity simulators, which can provide labeled training data. These algorithms are then applied to and calibrated with real data collected by the LHC experiments. The generated sets are small, ranging from 2 to 14 elements each, with around 0.9M sets divided to

train/val/test using the ratios 0.6/0.2/0.2. Each set element has 10 features ($d_{in}$). More information can be found in the supplementary material.

**Evaluation metrics, loss and post processing.** We consider multiple quantities to quantify the performance of the partitioning: the F1 score, the Rand Index (RI), and the Adjusted Rand Index (ARI $= (\text{RI} - \mathbb{E}[\text{RI}])/(1 - \mathbb{E}[\text{RI}]))$. All models are trained to minimize the F1 score. We make sure the adjacency matrix of the output graph encodes a valid partitioning of nodes to clusters by considering any connected components as a clique.

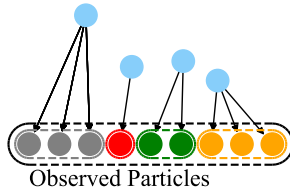

Observed Particles

Figure 4: Illustration of a particle physics experiment. The task is to partition the set of observed particles based on their point of origin (in blue).

**Results.** We compare the results of all learning based methods and a typical baseline algorithm used in particle physics (AVR). We also add the results of a trivial baseline that predicts that all nodes have the same progenitor. All models have roughly the same number of parameters. We performed each experiment 11 times with different random initializations, and evaluated the model F1 score, RI and ARI on the test set. The results are shown in Table 2-Jets. For bottom and charm jets, which have secondary vertices, both of our models significantly outperform the baselines by 5%-10% in all performance metrics. In light-jets, without secondary decays, our models yield similar scores. We also performed an extensive ablation study, see Table A1 in the supplementary material. Note that S2G+ has the same expressive power as S2G, and produces equivalent results in practice. Table 1 compares the training times of the different methods.

## 5.3 Learning Delaunay triangulations

In a second set-to-2-edge task we test our model's ability to learn Delaunay triangulations, namely given a set of planar points we want to predict the Delaunay edges between pairs of points, see e.g., [6] Chapter 9. We generated $50k$ planar point sets as training data and $5k$ planar point sets as test data; the point sets, $\boldsymbol{X} \in \mathbb{R}^{n \times 2}$, were uniformly sampled in the unit square, and a ground truth matrix in $\{0, 1\}^{n \times n}$ was computed per point set using a Delaunay triangulation algorithm. The number of points in a set, $n$, is either $50$ or varies and is randomly chosen from $\{20, \dots, 80\}$. Training was stopped after 100 epochs.

| Model | Epochs | training-time (minutes) |
|-------|--------|-------------------------|
| S2G   | 193    | 62  |
| S2G+  | 139    | 47  |
| GNN   | 91     | 21  |
| SIAM  | 77     | 24  |
| SIAM3 | 22     | 322 |
| MLP   | 132    | 22  |

Table 1: Training times of different models. Middle column: number of epochs with early stopping. Right column: total training time in minutes.

As in the previous experiment, all models have roughly the number of parameters. See more implementation details in the supplementary material. In Table 2-Delaunay we report accuracy of prediction as well as precision recall and F1 score. Evidently, both of our models (S2G and S2G+) outperform the baselines. We also tried the MLP baseline that yielded very low F1 scores (i.e., $\leq 0.1$). See also Figure 1 in the supplementary material for visualizations of several qualitative examples.

## 5.4 Set to 3-edges: learning the convex-hull of a point cloud

In the last experiment, we demonstrate learning of set-to-3-edge function. The learning task we target is finding supporting triangles in the convex hull of a set of points in $\mathbb{R}^3$. In this scenario, the input is a point set $\boldsymbol{X} \in \mathbb{R}^{n \times 3}$, and the function we wanted to learn is $\mathbf{F}^3 : \mathbb{R}^{n \times 3} \to \mathbb{R}^{n^3}$ where the output is a probability for each triplet of nodes (triangle) $\{\boldsymbol{x}_{i_1}, \boldsymbol{x}_{i_2}, \boldsymbol{x}_{i_3}\}$ to belong to the triangular mesh that describes the convex hull of $\boldsymbol{X}$. [35] had previously tackled a 2-dimensional version of this problem, but since their network predicts the order of nodes in the 2D convex hull, it is not easily adapted to the 3D settings.

Note that storing 3-rd order tensors in memory is not feasible, hence we concentrate on a *local* version of the problem: Given a point set $\boldsymbol{X} \subset \mathbb{R}^3$, identify the triangles within the $K$-Nearest-Neighbors of each point that belong to the convex hull of the entire point set $\boldsymbol{X}$. We used $K = 10$. Therefore, for broadcasting ($\boldsymbol{\beta}$) from point data to 3-edge data, instead of holding a 3-rd order tensor in memory we broadcast only the subset of $K$-NN neighborhoods. This allows working with high-order information with relatively low memory footprint. Furthermore, since we want to consider 3-edges (triangles)

| Model | F1 | RI | ARI | | Acc | Prec | Rec | F1 | | # points | F1 | AUC-ROC | | GT | predicted |
|---|---|---|---|---|---|---|---|---|---|---|---|---|---|---|---|
| | | | | | \multicolumn n = 50 | | | | | Spherical | | | | \multicolumn n = 30 | |

| Model | F1 | RI | ARI |
|---|---|---|---|
| **B** | | | |
| S2G | 0.646±0.003 | 0.736±0.004 | 0.491±0.006 |
| S2G+ | **0.655±0.004** | **0.747±0.006** | **0.508±0.007** |
| GNN | 0.586±0.003 | 0.661±0.004 | 0.381±0.005 |
| SIAM | 0.606±0.002 | 0.675±0.005 | 0.411±0.004 |
| SIAM-3 | 0.597±0.002 | 0.673±0.005 | 0.396±0.005 |
| MLP | 0.533±0.000 | 0.643±0.000 | 0.315±0.000 |
| AVR | 0.565 | 0.612 | 0.318 |
| trivial | 0.438 | 0.303 | 0.026 |
| **C** | | | |
| S2G | 0.747±0.001 | 0.727±0.003 | 0.457±0.004 |
| S2G+ | **0.751±0.002** | **0.733±0.003** | **0.467±0.005** |
| GNN | 0.720±0.002 | 0.689±0.003 | 0.390±0.005 |
| SIAM | 0.729±0.001 | 0.695±0.002 | 0.406±0.004 |
| SIAM-3 | 0.719±0.001 | 0.710±0.003 | 0.421±0.005 |
| MLP | 0.686±0.000 | 0.658±0.000 | 0.319±0.000 |
| trivial | 0.610 | 0.472 | 0.078 |
| AVR | 0.695 | 0.650 | 0.326 |
| **L** | | | |
| S2G | 0.972±0.001 | **0.970±0.001** | **0.931±0.003** |
| S2G+ | 0.971±0.002 | 0.969±0.002 | 0.929±0.003 |
| GNN | 0.972±0.001 | **0.970±0.001** | 0.929±0.003 |
| SIAM | **0.973±0.001** | **0.970±0.001** | 0.925±0.003 |
| SIAM-3 | 0.895±0.006 | 0.876±0.008 | 0.729±0.015 |
| MLP | 0.960±0.000 | 0.957±0.000 | 0.894±0.000 |
| trivial | 0.910 | 0.867 | 0.675 |
| AVR | 0.970 | 0.965 | 0.922 |

Jets

| | Acc | Prec | Rec | F1 |
|---|---|---|---|---|
| $n = 50$ | | | | |
| S2G | **0.984** | **0.927** | 0.926 | **0.926** |
| S2G+ | 0.983 | **0.927** | 0.925 | **0.926** |
| GNN0 | 0.826 | 0.384 | 0.966 | 0.549 |
| GNN5 | 0.809 | 0.363 | **0.985** | 0.530 |
| GNN10 | 0.759 | 0.311 | 0.978 | 0.471 |
| SIAM | 0.939 | 0.766 | 0.653 | 0.704 |
| SIAM-3 | 0.911 | 0.608 | 0.538 | 0.570 |
| $n \in \{20, \dots, 80\}$ | | | | |
| S2G | **0.947** | **0.736** | 0.934 | **0.799** |
| S2G+ | 0.947 | 0.735 | 0.934 | 0.798 |
| GNN0 | 0.810 | 0.387 | 0.946 | 0.536 |
| GNN5 | 0.777 | 0.352 | **0.975** | 0.506 |
| GNN10 | 0.746 | 0.322 | 0.970 | 0.474 |
| SIAM | 0.919 | 0.667 | 0.764 | 0.687 |
| SIAM-3 | 0.895 | 0.578 | 0.622 | 0.587 |

Delaunay

| | # points | F1 | AUC-ROC |
|---|---|---|---|
| Spherical | | | |
| S2G | 30 | **0.780** | **0.988** |
| GNN5 | 30 | 0.693 | 0.974 |
| SIAM | 30 | 0.425 | 0.885 |
| S2G | 50 | 0.686 | **0.975** |
| GNN5 | 50 | **0.688** | 0.973 |
| SIAM | 50 | 0.424 | 0.890 |
| S2G | 20-100 | 0.535 | 0.953 |
| GNN5 | 20-100 | **0.667** | **0.970** |
| SIAM | 20-100 | 0.354 | 0.885 |
| Gaussian | | | |
| S2G | 30 | **0.707** | **0.996** |
| GNN5 | 30 | 0.5826 | 0.9865 |
| SIAM | 30 | 0.275 | 0.946 |
| S2G | 50 | **0.661** | **0.997** |
| GNN5 | 50 | 0.4834 | 0.9917 |
| SIAM | 50 | 0.254 | 0.974 |
| S2G | 20-100 | **0.552** | **0.994** |
| GNN5 | 20-100 | 0.41 | 0.9866 |
| SIAM | 20-100 | 0.187 | 0.969 |

Convex Hull (a)

Convex Hull (b)

Table 2: Jets - Performance of partitioning for three types of jets. Delaunay - Results on the Delaunay triangulation task. Convex Hull (a) and (b) - Convex hull learning quantitative and qualitative results.

with no order we used invariant universal set model (DeepSets again) as $m$. For $k = 3$, **S2G** is implemented as follows: $\phi$ is implemented using DeepSets with 3 layers and output dimension $d_1 = 512$; $\beta$ triplets of points to sets. $\psi$ is implemented with a DeepSets with 3 layers of 64 features, followed by an MLP, $m$, with 3 layers. More details are in the supplementary.

We tested our S2G model on two types of data: Gaussian and spherical. For both types we draw point sets in $\mathbb{R}^3$ i.i.d. from standard normal distribution, $\mathcal{N}(0, 1)$, where for the spherical data we normalize each point to unit length. We generated $20k$ point set samples as a training set, $2k$ for validation and another $2k$ for test set. Point sets are in $\mathbb{R}^{n \times 3}$, where $n = 30$, $n = 50$, and $n \in [20, 100]$. We compare our method, S2G, to the SIAM, GNN and MLP baselines. The F1 scores and AUC-ROC of the predicted convex hull triangles are shown in Table 2-Convex Hull, where our model outperform the baselines in most cases (as in the previous experiment we exclude MLP from the table since it yields very low results). See Figure (b) for several examples of triangles predicted using our trained model compared to the ground truth.

## 5.5 Discussion

In the experiments above we compared our model S2G (and its variant S2G+) with several, broadly used, state-of-the-art architectures. As noted, our models compare favorably with these baselines in all tasks. We attribute this to the increased expressive power (universality) of these models. Our architectures are able to learn relations between set instances while reasoning about the whole set, where Siamese networks rely only on the relation between pairs or triplets of points. The GNN models use a prescribed connectivity that hinder efficient set learning.

## 6 Conclusion

In this paper, we presented a novel neural network family that can model equivariant set-to-$k$-edge functions and consequently set-to-graph, or more generally set-to-hypergraph functions. The family uses a relatively small number of parameters, compared to other equivariant models, and is shown to be a universal approximator of such continuous equivariant functions. We show the efficacy of these networks on several tasks, including a real-life particle physics problem. There are many directions for future work. One is adapting the model to learn valid clustering of sets (i.e., learn graphs that are built of disjoint cliques). Another direction is incorporating our network architecture in set models, with the goal of improving performance of general set tasks by constructing an intermediate latent graph representation, for example for constructing scene graphs as in [26].

## Broader Impact

Our contribution describes a class of neural network models for functions from sets to graphs and includes theoretical results that the model family is universal. The potential uses of these models is very broad and includes the physical sciences, computer graphics, and social networks. The paper includes experiments that show the positive impact of these models in the context of particle physics, and similar tasks appear repeatedly in the physical sciences. The models could also be used for social networks and areas with more complex ethical and societal consequences. Because the models treat the input as a set and are permutation equivariant, they have the potential to mitigate potential bias in data due to sorting and other pre-processing that could impact methods that treat the input as a sequence. Otherwise, the considerations of bias in the data and impact of failure are no different for our model than the generic considerations of the use of supervised learning and neural networks. Finally we note that the models we describe are already being used in real-world particle physics research.

## Acknowledgments and Disclosure of Funding

HS, NS and YL were supported in part by the European Research Council (ERC Consolidator Grant, "LiftMatch" 771136), the Israel Science Foundation (Grant No. 1830/17) and by a research grant from the Carolito Stiftung (WAIC). JS and EG were supported by the NSF-BSF Grant 2017600 and the ISF Grant 2871/19. KC was supported by the National Science Foundation under the awards ACI-1450310, OAC-1836650, and OAC-1841471 and by the Moore-Sloan data science environment at NYU.

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
