[Supplementary Material]

# Set2Graph: Learning Graphs From Sets: Supplementary Material

**Hadar Serviansky**[1]    **Nimrod Segol**[1]    **Jonathan Shlomi**[1]    **Kyle Cranmer**[2]

**Eilam Gross**[1]    **Haggai Maron**[3]    **Yaron Lipman**[1]

[1]Weizmann Institute of Science  [2]New York University  [3]NVIDIA Research

## 1   Architectures and hyper-parameters

Our **S2G** model (as well as **S2G+** for the main task) follows the formula $\mathbf{F}^k = \boldsymbol{\psi} \circ \boldsymbol{\beta} \circ \boldsymbol{\phi}$, where $\boldsymbol{\phi}$ is a set-to-set model, $\boldsymbol{\beta}$ is a *non-learnable* broadcasting set-to-graph layer, and $\boldsymbol{\psi}$ is a simple graph-to-graph network using only a single Multi-Layer Perceptron (MLP) acting on each $k$-edge feature vector independently. We note that all the hyper-parameters were chosen using the validation scores. All of the models used in the experiments are explained in section 5 in the main paper. Here, we add more implementation details, hyper-parameters and number of parameters.

**Notation.**   ”DeepSets / MLP of widths $[256, 256, 5]$” means that we use a DeepSets/MLP network with 3 layers, and each layer's output feature size is its corresponding argument in the array (e.g., the first and second layers have output feature size of $256$, while the third layer output feature size is $5$). Between the layers we use ReLU as a non linearity.

**Partitioning for particle physics applications.**   For our model **S2G**, $\boldsymbol{\phi}$ is implemented using DeepSets Zaheer et al. (2017) with 5 layers of width $[256, 256, 256, 256, 5]$. $\boldsymbol{\psi}$ is implemented with an MLP $[256, 1]$, with output considered as edge probabilities. Instead of using a max or sum pooling in DeepSets layers, we used a self-attention mechanism based on Ilse et al. (2018) and Vaswani et al. (2017):

$$Attention\left(\boldsymbol{X}\right) = \text{softmax}\left(\frac{\tanh f_1(\boldsymbol{X}) \cdot f_2(\boldsymbol{X})^T}{\sqrt{d_{\text{small}}}}\right) \cdot \boldsymbol{X}, \tag{A1}$$

where $f_1, f_2$ are implemented by two single layer MLPs of width $\left[d_{\text{small}} = \frac{d_X}{10}\right]$. The model has 457449 learnable parameters. **S2G+** is identical to **S2G**, except that $\boldsymbol{\beta}$ is defined using the full equivariant basis $\mathbb{R}^n \to \mathbb{R}^{n^2}$ from Maron et al. (2019) that contains $\text{bell}(3) = 5$ basis operations. It has 461289 learnable parameters.

We used a grid search for the following hyper-parameters: learning rate in

$$\{1e-5, 3e-5, 1e-4, 3e-4, 1e-3, 3e-3\},$$

DeepSets layers' width in $\{64, 128, 256, 512\}$, number of layers in $\{3, 4, 5\}$, $\boldsymbol{\psi}$ (MLP) with widths in $\{[128, 1], [256, 1], [512, 1], [128, 256, 128, 1]\}$, and with or without attention mechanism in DeepSets.

The following hyper-parameters choice is true for all models, unless stated otherwise. As a loss, we used a combination of soft F1 score loss and an edge-wise binary cross-entropy loss. We use early stopping based on validation score, batch size of 2048, adam optimizer Kingma and Ba (2014) with learning rate of $1e-3$. Training takes place in less than 2 hours on a single Tesla V100 GPU.

The deep learning baselines are implemented as follows: **SIAM** is implemented similarly to S2G, with the exception that instead of using DeepSets as $\phi$, we use MLP $[384, 384, 384, 384, 5]$. The learning rate is $3e-3$ and SIAM has 452742 learnable parameters. **SIAM-3** uses a Siamese MLP of widths $[384, 384, 384, 384, 20]$ to extract node features, and the edge logits are the $l_2$ distances between the nodes. SIAM-3 uses a triplets loss Weinberger et al. (2006) - we draw random triplets *anchor, neg, pos* where *anchor* and *pos* are of the same cluster, and *neg* is of a different cluster, and the loss is defined as [1]

$$L_i = \min \left( d_{l2} \left( anch_i, pos_i \right) - d_{l2} \left( anch_i, neg_i \right) + 2, 0 \right)$$

The learning rate is $1e-4$ and SIAM-3 has 455444 learnable parameters. Due to the triplet choice process, training takes place in around 9 hours. **MLP** is a straight-forward fully-connected network acting on the flattened feature vectors of the input sets. It uses fully-connected layers of widths $\left[ 512, 256, 512, 15^2 \right]$, and has 455649 learnable parameters. **GNN** is a GraphConv network Morris et al. (2018) where the underlying graph is selected as the $k$-NN ($k = 5$) graph constructed using the $l_2$ distance between the elements' feature vectors. the GraphConv layers have $[350, 350, 300, 20]$ features, and the edge logits between set elements are based on the inner product between the features of each 2 elements. GNN has 455900 learnable parameters.

The dataset is made of training, validation and test set with 543544, 181181 and 181182 instances accordingly. Each of the sets contains all three flavors: bottom, charm and light jets roughly in the same amount, while the flavor of each instance is not part of the input. We repeat the following evaluation 11 times: (1) training over the dataset, early-stopping when the F1 score over the validation set does not improve for 20 epochs. (2) Predicting the clusters of the test set. (3) Separate the 3 flavors and calculate the metrics for each flavor. Eventually, we have 11 scores for each combination of metrics, flavor and model, and we report the mean±std. Note that the AVR is evaluated only once since it is not a stochastic algorithm. We also conducted an ablation study for this experiment, the results for all particle types can be found in Table A1. Examples of inferred graphs can be seen in Figure A1. Since edges are predicted, there's a need to project the inferred graph to a connected-components graph. The results show that in most cases, the inferred graphs are already predicted that way.

**Learning Delaunay triangulation.** In our model **S2G, S2G+**, $\phi$ is an MLP $[500, 500, 500, 1000, 500, 500, 80]$. $\beta$ is broadcasting as before for models S2G and S2G+, thus ending with 160 or 400 features per edge, respectively. $\psi$ is an MLP $[1000, 1000, 1]$, ending as the edge probability. We use edge-wise binary cross-entropy loss. **S2G** and **S2G+** have 5918742 and 6398742 learnable parameters respectively. The implementation of **SIAM** baseline is as follows: $\phi$ is an MLP $[700, 700, 700, 1400, 700, 700, 112]$, $\beta$ is broadcasting as in S2G, and $\psi$ is MLP $[1400, 1400, 1]$ and edge-wise binary cross-entropy loss. **SIAM** has 5804037 learnable parameters. **SIAM-3** uses a Siamese MLP $[500, 1000, 1500, 1250, 1000, 500, 500, 80]$ and has 5922330 learnable parameters. **GNN**$k$ is as the previous experiment, where $k \in \{0, 5, 10\}$, with 3 GraphConv layers of widths $[1000, 1500, 1000]$, and it has 6007500 learnable parameters. We searched learning rate from $\{1e-2, 1e-3, 1e-4\}$, using $1e-3$ with Adam optimizer. All models trained for up to 8 hours on a single Tesla V100 GPU. A qualititive result is shown in Figure A2.

**Set to 3-edges.** For **S2G**, $\phi$ is implemented using DeepSets $[512, 512, 512]$. In this task, the model predicts supporting triangles of the convex hull, also referred to as faces, among triplets in the KNN graph. Hence, we do not maintain 3-rd order tensors in the memory. For each node we aggregate all the triangles which lie in its KNN ($k = 10$) neighbors. In order to be invariant to the order of the 3 nodes in a face (i.e., the output tensor is symmetric w.r.t. permutations of the triplets' order), each triplets is viewed as a set and fed to a DeepSets $[64, 64, 64]$, max-pooled, and then to an MLP of widths $[256, 128, 1]$. The model has 1186689 learnable parameters. As a loss, we used a combination of soft F1 score loss and a face-wise binary cross-entropy loss. **SIAM** is identical except that $\phi$ is implemented by MLP $[1024, 1024, 512]$, and the second DeepSets is replaced by an MLP $[128, 128, 64]$. It has 1718593 learnable parameters. **GNN5** is implemented as in the first experiments, with $k = 5$, GraphConv layers $[512, 512, 512, 128]$ and the hyper-edge logits are computed as the sum of the product between the corresponding 3 nodes. It has 1184384 learnable paramaters.

| | Method | $\psi$ #layers | $\phi$ | $\phi$ #layers | $d_1$ | Attention | F1 | RI | ARI |
|---|---|---|---|---|---|---|---|---|---|
| b jets | S2G | 2 | DeepSets | 5 | 2 | V | 0.649 | 0.736 | 0.493 |
| | S2G | 2 | DeepSets | 5 | 10 | V | 0.642 | 0.739 | 0.488 |
| | S2G+ | 2 | DeepSets | 5 | 5 | V | 0.658 | 0.745 | 0.510 |
| | S2G | 2 | Siamese | 5 | 5 | V | 0.605 | 0.671 | 0.408 |
| | S2G+ | 2 | DeepSets | 4 | 5 | V | 0.649 | 0.733 | 0.493 |
| | S2G+ | 2 | DeepSets | 5 | 2 | V | 0.642 | 0.732 | 0.484 |
| | S2G+ | 2 | DeepSets | 6 | 5 | V | 0.654 | 0.739 | 0.502 |
| | S2G | 2 | DeepSets | 5 | 5 | X | 0.640 | 0.726 | 0.478 |
| | S2G | 2 | DeepSets | 4 | 5 | V | 0.649 | 0.741 | 0.498 |
| | S2G | 2 | PointNetSeg | 5 | 5 | V | 0.630 | 0.720 | 0.462 |
| | S2G | 2 | DeepSets | 5 | 5 | V | 0.646 | 0.739 | 0.495 |
| | QUAD | 1 | DeepSets | 5 | 5 | V | 0.637 | 0.730 | 0.470 |
| | S2G+ | 2 | DeepSets | 5 | 10 | V | 0.655 | 0.749 | 0.510 |
| | S2G | 2 | DeepSets | 6 | 5 | V | **0.660** | **0.753** | **0.516** |
| | S2G+ | 2 | Siamese | 5 | 5 | V | 0.438 | 0.303 | 0.026 |
| | S2G+ | 2 | DeepSets | 5 | 5 | X | 0.643 | 0.729 | 0.482 |
| | S2G | 1 | DeepSets | 5 | 5 | V | 0.565 | 0.710 | 0.395 |
| | S2G+ | 2 | PointNetSeg | 5 | 5 | V | 0.619 | 0.717 | 0.451 |
| | S2G+ | 1 | DeepSets | 5 | 5 | V | 0.577 | 0.717 | 0.414 |
| c jets | S2G | 2 | DeepSets | 5 | 2 | V | 0.749 | 0.727 | 0.458 |
| | S2G | 2 | DeepSets | 5 | 10 | V | 0.747 | 0.729 | 0.459 |
| | S2G+ | 2 | DeepSets | 5 | 5 | V | 0.753 | 0.732 | 0.467 |
| | S2G | 2 | Siamese | 5 | 5 | V | 0.728 | 0.693 | 0.404 |
| | S2G+ | 2 | DeepSets | 4 | 5 | V | 0.748 | 0.726 | 0.456 |
| | S2G+ | 2 | DeepSets | 5 | 2 | V | 0.749 | 0.726 | 0.457 |
| | S2G+ | 2 | DeepSets | 6 | 5 | V | 0.750 | 0.729 | 0.462 |
| | S2G | 2 | DeepSets | 5 | 5 | X | 0.743 | 0.720 | 0.444 |
| | S2G | 2 | DeepSets | 4 | 5 | V | 0.749 | 0.728 | 0.460 |
| | S2G | 2 | PointNetSeg | 5 | 5 | V | 0.741 | 0.720 | 0.443 |
| | S2G | 2 | DeepSets | 5 | 5 | V | 0.750 | 0.730 | 0.463 |
| | QUAD | 1 | DeepSets | 5 | 5 | V | 0.750 | 0.734 | 0.469 |
| | S2G+ | 2 | DeepSets | 5 | 10 | V | 0.752 | **0.735** | **0.470** |
| | S2G | 2 | DeepSets | 6 | 5 | V | **0.754** | 0.734 | 0.470 |
| | S2G+ | 2 | Siamese | 5 | 5 | V | 0.610 | 0.472 | 0.078 |
| | S2G+ | 2 | DeepSets | 5 | 5 | X | 0.741 | 0.718 | 0.439 |
| | S2G | 1 | DeepSets | 5 | 5 | V | 0.699 | 0.694 | 0.383 |
| | S2G+ | 2 | PointNetSeg | 5 | 5 | V | 0.738 | 0.718 | 0.440 |
| | S2G+ | 1 | DeepSets | 5 | 5 | V | 0.705 | 0.701 | 0.394 |
| light jets | S2G | 2 | DeepSets | 5 | 2 | V | 0.973 | 0.971 | 0.933 |
| | S2G | 2 | DeepSets | 5 | 10 | V | 0.970 | 0.968 | 0.927 |
| | S2G+ | 2 | DeepSets | 5 | 5 | V | 0.973 | 0.970 | 0.932 |
| | S2G | 2 | Siamese | 5 | 5 | V | 0.973 | 0.970 | 0.926 |
| | S2G+ | 2 | DeepSets | 4 | 5 | V | 0.973 | 0.971 | 0.933 |
| | S2G+ | 2 | DeepSets | 5 | 2 | V | 0.974 | **0.972** | **0.935** |
| | S2G+ | 2 | DeepSets | 6 | 5 | V | 0.972 | 0.970 | 0.931 |
| | S2G | 2 | DeepSets | 5 | 5 | X | 0.973 | 0.971 | 0.931 |
| | S2G | 2 | DeepSets | 4 | 5 | V | 0.972 | 0.970 | 0.930 |
| | S2G | 2 | PointNetSeg | 5 | 5 | V | **0.974** | 0.971 | 0.933 |
| | S2G | 2 | DeepSets | 5 | 5 | V | 0.972 | 0.970 | 0.931 |
| | QUAD | 1 | DeepSets | 5 | 5 | V | 0.972 | 0.970 | 0.929 |
| | S2G+ | 2 | DeepSets | 5 | 10 | V | 0.970 | 0.968 | 0.928 |
| | S2G | 2 | DeepSets | 6 | 5 | V | 0.972 | 0.971 | 0.932 |
| | S2G+ | 2 | Siamese | 5 | 5 | V | 0.910 | 0.867 | 0.675 |
| | S2G+ | 2 | DeepSets | 5 | 5 | X | 0.973 | 0.971 | 0.933 |
| | S2G | 1 | DeepSets | 5 | 5 | V | 0.968 | 0.969 | 0.926 |
| | S2G+ | 2 | PointNetSeg | 5 | 5 | V | 0.973 | 0.972 | 0.934 |
| | S2G+ | 1 | DeepSets | 5 | 5 | V | 0.966 | 0.967 | 0.923 |

Table A1: Ablation study for particle partitioning.

Figure A1: Jets qualitative results. For each pair, the left side is before completing edges to a connected-component graph. The color of the vertices refer to the GT cluster. The edges are predicted by the model.

Figure A2: Results of Delaunay triangulation learning. Top: $n = 50$; Bottom: $n \in \{20, \ldots, 80\}$.

For hyper-parameters search, we examined learning rates in

$$\{1e-5, 3e-5, 1e-4, 3e-4, 1e-3, 3e-3\},$$

and DeepSets models of width $\{64, 128, 256, 512\}$. We used Adam optimizers with learning rate of $1e-3$. Training took place for up to 36 hours on a single Tesla V100 GPU.

## 2    Proofs

*Proof (of Theorem 3).* The general proof idea is to consider an arbitrary equivariant set-to-$k$-edge polynomial $\mathbf{P}^k$ and use its equivariance property to show that it has the form as in equation 6. This is done by looking at a particular output entry $\mathbf{P}^k_{i^0}$, where say $\boldsymbol{i}^0 = (1, 2, \ldots, k)$. Then the proof considers two subsets of permutations: First, the subgroup of all permutations $\sigma \in S_n$ that fixes the numbers $1, 2, \ldots, k$, i.e., $\sigma(\boldsymbol{i}^0) = \boldsymbol{i}^0$, but permute everything else freely; this subgroup is denoted $\mathrm{stab}(\boldsymbol{i}^0)$. Second, permutations of the form $\sigma = (1 \; i_1)(2 \; i_2) \cdots (k \; i_k)$, where $\boldsymbol{i} \in [n]^k$. Each of these permutation subsets reveals a different part in the structure of the equivariant polynomial $\mathbf{P}^k$ and its relation to invariant polynomials. We provide the full proof next.

It is enough to prove Theorem 3 for $d_{\mathrm{out}} = 1$. Let $\boldsymbol{i}^0 = (1, 2, \ldots, k)$ and consider any permutation $\sigma \in \mathrm{stab}(\boldsymbol{i}^0)$. Then from equivariance of $\mathbf{P}^k$ we have

$$\mathbf{P}^k_{\boldsymbol{i}^0}(\boldsymbol{X}) = \mathbf{P}^k_{\sigma^{-1}(\boldsymbol{i}^0)}(\boldsymbol{X}) = \mathbf{P}^k_{\boldsymbol{i}^0}(\sigma \cdot \boldsymbol{X}),$$

and $\sigma \cdot \boldsymbol{X} = (\boldsymbol{x}_1, \ldots, \boldsymbol{x}_k, \boldsymbol{x}_{\sigma^{-1}(k+1)}, \ldots, \boldsymbol{x}_{\sigma^{-1}(n)})^T$. That is $\mathbf{P}^k_{\boldsymbol{i}^0}$ is invariant to permuting its last $n-k$ elements $\boldsymbol{x}_{k+1}, \ldots, \boldsymbol{x}_n$; we say that $\mathbf{P}_{\boldsymbol{i}^0}$ is $S_{n-k}$ invariant. We next prove that $S_{n-k}$ invariance can be written using a combination of $S_n$ invariant polynomials and tensor products of $\boldsymbol{x}_1, \ldots, \boldsymbol{x}_k$:

**Lemma A1.** *Let $p(\boldsymbol{X}) = p(\boldsymbol{x}_1, \ldots, \boldsymbol{x}_k, \boldsymbol{x}_{k+1}, \ldots, \boldsymbol{x}_n)$ be $S_{n-k}$ invariant polynomial. That is invariant to permuting the last $n-k$ terms. Then*

$$p(\boldsymbol{X}) = \sum_{\boldsymbol{\alpha}} \boldsymbol{x}_1^{\alpha^1} \cdots \boldsymbol{x}_k^{\alpha^k} q_{\boldsymbol{\alpha}}(\boldsymbol{X}), \tag{A2}$$

*where $q_{\boldsymbol{\alpha}}$ are $S_n$ invariant polynomials.*

Before we provide the proof of this lemma let us finish the proof of Theorem 3. So now we know that $\mathbf{P}^k_{\boldsymbol{i}^0}$ has the form equation A2. On the other hand let $\boldsymbol{i}$ be an arbitrary multi-index and consider the permutation $\sigma = (1 \; i_1)(2 \; i_2) \cdots (k \; i_k)$. Again by permutation equivariance of $\mathbf{P}^k$ we have

$$\mathbf{P}^k_{i_1 i_2 \cdots i_k}(\boldsymbol{X}) = \mathbf{P}^k_{\sigma^{-1}(\boldsymbol{i}_0)}(\boldsymbol{X}) = \mathbf{P}^k_{\boldsymbol{i}_0}(\sigma \cdot \boldsymbol{X})$$
$$= \sum_{\boldsymbol{\alpha}} \boldsymbol{x}_{i_1}^{\alpha^1} \cdots \boldsymbol{x}_{i_k}^{\alpha^k} q_{\boldsymbol{\alpha}}(\boldsymbol{X}),$$

which is a coordinate-wise form of equation 6 with $d_{\mathrm{out}} = 1$. ☐

*Proof (of Lemma A1).* First we expand $p$ as

$$p(\boldsymbol{X}) = \sum_{\boldsymbol{\alpha}} \boldsymbol{x}_1^{\alpha^1} \cdots \boldsymbol{x}_k^{\alpha^k} q_{\boldsymbol{\alpha}}(\boldsymbol{x}_{k+1}, \ldots, \boldsymbol{x}_n), \tag{A3}$$

where $p_{\boldsymbol{\alpha}}$ are $S_{n-k}$ invariant polynomials. Since $S_{n-k}$ invariant polynomials with variables $\boldsymbol{x}_{k+1}, \ldots, \boldsymbol{x}_n$ are generated by the power-sum multi-symmetric polynomials

$$\sum_{i=k+1}^n \boldsymbol{x}_i^\alpha = \sum_{i=1}^n \boldsymbol{x}_i^\alpha - \sum_{i=1}^k \boldsymbol{x}_i^\alpha,$$

with $|\alpha| \leq n - k$, see e.g., Rydh (2007), we have that each $p_{\boldsymbol{\alpha}}(\boldsymbol{x}_{k+1}, \ldots, \boldsymbol{x}_n) = \sum_{\boldsymbol{\alpha}} \boldsymbol{x}_1^{\alpha^1} \cdots \boldsymbol{x}_k^{\alpha^k} r_{\boldsymbol{\alpha}}(\boldsymbol{X})$, for some $S_n$ invariant polynomials $r_{\boldsymbol{\alpha}}$. Plugging this in equation A3 proves the lemma.

$\square$

*Proof (Lemma 1).* We can assume $d_{\text{out}} = 1$. The general case is proved by finding approximating polynomial to each output feature coordinate. Let $\epsilon > 0$. Using Stone-Weierstrass we can find a polynomial $\mathbf{Q} : K \to \mathbb{R}^{n^k}$ so that $\max_{\boldsymbol{X} \in K} \left\| \mathbf{G}^k(\boldsymbol{X}) - \mathbf{Q}(\boldsymbol{X}) \right\|_\infty < \epsilon$. Let

$$\mathbf{P}^k(\boldsymbol{X}) = \frac{1}{n!} \sum_{\sigma \in S_n} \sigma \cdot \mathbf{Q}(\sigma^{-1} \cdot \boldsymbol{X}).$$

Then $\mathbf{P}^k$ is equivariant and since $\mathbf{G}^k$ is also equivariant we have

$$\left\| \mathbf{G}^k(\boldsymbol{X}) - \mathbf{P}^k(\boldsymbol{X}) \right\|_\infty$$
$$= \frac{1}{n!} \left\| \sum_{\sigma \in S_n} \sigma \cdot \left( \mathbf{G}^k(\sigma^{-1} \cdot \boldsymbol{X}) - \mathbf{Q}(\sigma^{-1} \cdot \boldsymbol{X}) \right) \right\|_\infty$$
$$< \frac{1}{n!} \sum_{\sigma \in S_n} \epsilon = \epsilon.$$

$\square$

**Approximating $\mathbf{P}^k$ with a network $\mathbf{F}^k$.** We set a target $\epsilon > 0$. Let $U \supset \boldsymbol{H}(K)$ be a compact $\epsilon$-neighborhood of $\boldsymbol{H}(K)$. $p$ is uniformly continuous over $\cup_i \boldsymbol{\beta}(U)_{i,:}$. Choose $\boldsymbol{m}$ so to be an $\epsilon/2$-approximation to $p$ over $\cup_i \boldsymbol{\beta}(U)_{i,:}$. Let $\delta$ be so that for $\boldsymbol{Y}, \boldsymbol{Y}' \in U$, $\|\boldsymbol{Y} - \boldsymbol{Y}'\|_\infty < \delta$ implies $\|p(\boldsymbol{\beta}(\boldsymbol{Y})) - p(\boldsymbol{\beta}(\boldsymbol{Y}'))\|_\infty < \epsilon/2$. Now choose $\phi$ so that it is $\delta_0$-approximation to $\boldsymbol{H}$ over $K$ where $\delta_0 < \min \{\delta, \epsilon\}$. This can be done since $\phi$ is a universal set-to-set model as in Segol and Lipman (2020). Lastly, putting all the pieces together we get for all $\boldsymbol{i}$:

$$|p(\boldsymbol{\beta}(\boldsymbol{H}(\boldsymbol{X}))_{\boldsymbol{i},:}) - \boldsymbol{m}(\boldsymbol{\beta}(\phi(\boldsymbol{X}))_{\boldsymbol{i},:})| \leq$$
$$|p(\boldsymbol{\beta}(\boldsymbol{H}(\boldsymbol{X}))_{\boldsymbol{i},:}) - p(\boldsymbol{\beta}(\phi(\boldsymbol{X}))_{\boldsymbol{i},:})| +$$
$$|p(\boldsymbol{\beta}(\phi(\boldsymbol{X}))_{\boldsymbol{i},:}) - \boldsymbol{m}(\boldsymbol{\beta}(\phi(\boldsymbol{X}))_{\boldsymbol{i},:})| < \epsilon.$$

*Proof (Proposition 1).* Consider the case $k = 2$ and the constant function set-to-graph function $\mathbf{G}(\boldsymbol{X}) = \boldsymbol{I}$, where $\boldsymbol{I}$ is the identity $n \times n$ matrix; that is $\mathbf{G}$ learns the constant value 1 for 1-edges (nodes), and 0 for 2-edges. Since $\phi$ is equivariant we have that $\phi(\mathbf{1}) = \mathbf{1} \otimes \boldsymbol{a} = \mathbf{1}\boldsymbol{a}^T$, for some vector $\boldsymbol{a} \in \mathbb{R}^{d_1}$. Therefore $\boldsymbol{\beta}(\phi(\mathbf{1}))_{i_1, i_2, :} = [\boldsymbol{a}, \boldsymbol{a}]$ and $\boldsymbol{m}(\boldsymbol{\beta}(\phi(\mathbf{1}))) = \boldsymbol{m}([\boldsymbol{a}, \boldsymbol{a}]) = b \in \mathbb{R}$. We get that $\mathbf{F}^2(\mathbf{1})_{i_1, i_2, :} = b$ and $\left\| \boldsymbol{I} - \mathbf{F}^2(\mathbf{1}) \right\|_\infty \geq 1/2$.  $\square$

## 3   Physics background.

The Large Hadron Collider (LHC) is the world's highest energy particle collider, located at the CERN laboratory in Geneva, and is used to study the fundamental particles of nature and their interactions.

Figure A3: Distribution of the number of partitions in each type of set.

The LHC collides proton beams at high energy, and these collisions produce many new particles, which may be unstable or lead to subsequent particle production. For instance, the production of quarks (fundamental particles that make up protons, neutrons, and other hadrons) will lead to the production of many hadrons and eventually be manifest as a spray of particles called a *jet*. The collisions take place in a vacuum, but the collision point is surrounded by large detectors that measure the outgoing particles that are stable enough to reach the detector several centimeters away. In order to probe the properties of particles that are unstable, we need to infer which "flavor" of quark was the progenitor particle that led to a jet. This classification is performed in many stages, and we focus on a particular aspect of it known as vertex reconstruction, which we describe below.

The location of the initial proton-proton collision is referred to as the primary vertex. Several particles emanating from the primary vertex are stable, will reach the detector, and will be part of the observed jet. Other particles will be unstable and travel a few millimeters before decaying in what is referred to as a secondary vertex. The unstable particles are not observed; however, the trajectories of the stable charged particles will be measured by detectors located around the collision point. Due to the presence of magnetic fields in the detector, the trajectories of the charged particles are helical. The helical trajectories are called *tracks*, and are summarized by 6 numbers, called the perigee parameters, along with a covariance matrix quantifying the uncertainty of their measurement.

Vertex reconstruction can be composed into two parts, vertex finding and vertex fitting. Vertex finding refers to partitioning the tracks into vertices, and vertex fitting is computing the most likely origin point in space for a collection of tracks. In the standard vertex reconstruction algorithms, these two parts are often intertwined and done together. In this application we perform the partitioning without performing the actual geometrical fit. From the physics point of view, once we improve the partitioning, the identification of the jets flavor is naturally improved. Vertex reconstruction propagates to a number of down-stream data analysis tasks, such as particle identification (a classification problem). Therefore, improvements to the vertex reconstruction has significant impact on the sensitivity of collider experiments.

**Dataset.** Algorithms for particle physics are typically designed with high-fidelity simulators, which can provide labeled training data. These algorithms are then applied to and calibrated with real data collected by the LHC experiments. Our simulated samples are created with a standard simulation package called PYTHIA Sjöstrand et al. (2015) and the detector is simulated with DELPHES de Favereau et al. (2014). We use this software to generate synthetic datasets for three types (called "flavors") of jets. The generated sets are small, ranging from 2 to 14 elements each. The three different jet types are labeled bottom-jets, charm-jets, and light-jets (B/C/L). The important distinction between the flavors is the typical number of partitions in each set. Figure A3 shows the distribution of the number of partitions (vertices) in each flavor: bottom jets typically have multiple partitions; charm jets also have multiple partitions, but fewer than bottom jets; and light jets typically have only one partition.

**AVR algorithm.** We compare the model performance to a non-learning algorithm, (AVR), implemented in RAVE Waltenberger (2011). The basic concept of AVR is to perform a least squares fit of the vertex position given the track trajectories and their errors, remove less compatible tracks from the fit, and refit them to secondary vertices.

## Footnotes

[1] A natural disadvantage of the triplets loss is that it cannot learn from sets with a single cluster, or sets with size 2.