[Reviews · NeurIPS 2020]

Review 1

Summary and Contributions: ------------------------------------------------------------------- Post Rebuttal: After reading the response from the authors, I am sufficiently satisfied with majority of my concerns being addressed and the addition of the two tables to the revised version. My assessment remains positive. ------------------------------------------------------------------- The authors propose a framework to learn functions to output graphs/ hypergraphs (including non-uniform hypergraphs) with input as sets derived from principles which preserve equivariance. They provide universality results for the framework (using functions F^1 through to F^k where each F^i is able to represent edge incident on 'i' vertices) for the same with the help of a model family with 3 constituent functions (one function which is a non learnable broadcasting function) applied sequentially to the input set. They provide results on 3 datasets (1 physics simulation dataset and 2 synthetic datasets) to validate the empirical benefits of their proposed model over models from related areas (namely similarity learning).

Strengths: Set/ Graph/ Hypergraph Representation Learning models with equivariance as an inductive bias are of growing interest to the community. More concretely constructing graphs, hypergraphs from an input set has been an important problem in machine learning and this work provides a theoretically sound procedure to build the same. The work provides a learning framework with universality results without the explicit requirement of higher order tensors with high memory footprint - and a practical model based on the theory - which showcases the positives of the model.

Weaknesses: 1. While the framework by itself doesn't require higher order tensors, tensor scalability issues still remain and the method may not be best suited computationally when there is high sparsity in the output graphs, hypergraphs - where an incident structure is a more compact representation (the broadcasting function employed doesn't directly allow for this to be done). Also, please provide the running time for each of the baseline models in comparison to the proposed model. 2. While the baselines themselves have good coverage, the experimental datasets do not showcase the power of the model. For example, in the physics dataset, a trivial predictor does pretty well in terms of F1 and R1 scores. 3. The authors employ an MLP towards the psi component and treat each hyperedge - edge to be independent of the others - while they are still universal - this may not be sufficient for more complex datasets and appears like the complexity is pushed to the earlier layers of the network (phi component).

Correctness: To the best of my knowledge, the proof of the theorems and the lemmas are complete with reasonable assumptions.

Clarity: Yes, the paper is well written - especially in terms of the way the theorems 1 and 2 are proved using a three step procedure. However, the paper can be made more reader friendly if the authors could add a figure for section 4.2 which could explain the setup towards proving theorem 3 (which is ultimately used for theorems 1 and 2).

Relation to Prior Work: To the best of my knowledge, the related work pertaining to the theoretical contributions of the paper are comprehensive.

Reproducibility: Yes

Additional Feedback: Please address the concerns presented in the weaknesses section or highlight them in the paper - as best possible.


Review 2

Summary and Contributions: This paper proposed a family of neural networks that can map a set to graph edges. The algorithm is evaluated on particle physics.

Strengths: It is uncommon to model the problem as set2graph. Therefore, I would say this paper provides novelty.

Weaknesses: I don't see how the set2graph formulation is clearly justified. For example, the author mentioned many ML problems can be categorized as set2graph like clustering. However, I don't see why classical clustering algorithms cannot work. There are even more candidates than clustering, e.g., pair-wise learning to rank. Given no clear motivation of the s2g formulation, I hope to see that it is justified by empirical evaluation. However, it is also not clear. For example, for the GNN baseline, there are tons of link prediction models in addition to the standard GCN. The author didn't give clear reason why those won't work. As a result, I'm not sure if the results are valuable.

Correctness: Baselines are not sufficient.

Clarity: The writing can be significantly improved. For example, the title can be changed to something specific to a particular application much smaller than general set2graph. The mentioned "clustering, learning vertex and edge features on graphs, and learning features on triplets" scope are not well justified. It's kinda of strange to do line changing in abstract.

Relation to Prior Work: It is not clear how classical methods like clustering or recent advances in GNN won't work.

Reproducibility: Yes

Additional Feedback:


Review 3

Summary and Contributions: This paper proposes a way to construct equivalent set-to-hypergraph functions by composing three functions, \phi, \beta, and \psi. The paper also shows the universality; the class of functions can approximate arbitrary equivalent set-to-hypergraph functions. Before this research, there was no practically feasible equivalent set-to-hypergraph functions with universality guarantee.

Strengths: The proposed function class is practical, and it could be applied to popular tasks such as clustering. It comes with universality guarantee and good experimental results. This type of theoretical soundness is celebrated in NeurIPS community.

Weaknesses: The author could compare the computational complexity, or the actual computational time of the models in the experiments.

Correctness: I followed the proofs in the paper, and as far as I can tell, they are correct.

Clarity: The paper is very well written. The author sometimes shows k=2 case as an example, which helped me understand the paper a lot.

Relation to Prior Work: The relation to the prior works are mentioned in section 2.

Reproducibility: Yes

Additional Feedback: Some point it didn't clear to me was * Is \phi (from DeepSets) guaranteed to be set-to-set equivalent? * How should we set the loss function? For k=2 case, should we back propagate from n*(n-1)/2 losses or n*(n-1)?


Review 4

Summary and Contributions: I'm satisfied with the authors' rebuttal and my decision remains the same. --------------------------------------------------------------- This paper targets at Set2Graph problem. The authors show that low order tensors are sufficient for set-to-graph universality and build an equivariant model for the set-to-graph problem which is both practical and provably universal.

Strengths: The universality of the propsoed model is proven. The introduction of the proposed method is clear. The experiments are sufficient (on various datasets and with differnt settings).

Weaknesses: My main concern is the scalability of the proposed method. The broadcasting operation will create a n^k * d_2 tensor, which seems imply that the proposed model is inapplicable to large-scale graphs (i.e., n is large) or high-order graphs (i.e., k > 3). It would be nice if authors can show the time and space complexity of the method and discuss the computational limitations of the proposed model. Or, maybe the authors would like to discuss the potential solution to mitigate this issue. Additionally, I suggest to introduce some learnable operations in beta, which may be beneficial for the scalability of the method, e.g., for each modality of the tensor, apply a linear/nonlinear map to reduce the dimensionality of x. Maybe the authors can show some experiments for such a variant. Compared with S2G, S2G+ does not show obvious improvements. Could authors provide more analysis on this phenomenon? Could the authors add some visualization results for the generated graphs based on typical input sets, and discuss the rationality of the proposed method based on the generated graphs? Such analytic content will be beneficial for us to understand the mechanism of the proposed method. Besides universality, could the authors discuss the permutation-invariance (or permutation-equivariance) of the proposed method, which I think is important for many point set-driven applications? If the input point set contains some noise or outliers, will the representations achieved by the proposed method be robust? I would like to see some robustness analysis and experiments. Minors: the figures in the paper should be enlarged.

Correctness: Yes.

Clarity: Yes.

Relation to Prior Work: Yes.

Reproducibility: Yes

Additional Feedback:

[Author Response · NeurIPS 2020]

We thank the reviewers for their thoughtful comments and suggestions. We will incorporate them in our revised version. Below, we address the main questions and concerns that were raised in the reviews.

**(R1, R3, R4) "...compare the computational complexity, or the actual computational time of the models..."** This is a great suggestion. Table 1 compares the training time for all of the models on the particle physics experiment. All models ran on the same hardware. Stopping criteria is after 20 epochs with no f1-score improvement. Table 2 presents the computational complexity analysis of our method **S2G** for the graph ($k = 2$) case in two scenarios: sparse and dense tensors. We assume that the S2G model represents a $f : \mathbb{R}^{n \times d} \to \mathbb{R}^{n^2}$ function, and that the feature dimension is constant across all layers (e.g., taking it to be the maximum across all layers). The S2G model is composed of the following functions: $\phi : \mathbb{R}^{n \times d} \to \mathbb{R}^{n \times d}, \beta : \mathbb{R}^{n \times d} \to \mathbb{R}^{n^2 \times 2d}, \psi : \mathbb{R}^{n^2 \times 2d} \to \mathbb{R}^{n^2}$. We will add both tables to the final version.

| Model | Epochs | Run-time (minutes) |
|---|---|---|
| S2G | 193 | 62 |
| S2G+ | 139 | 47 |
| GNN | 91 | 21 |
| SIAM | 77 | 24 |
| SIAM3 | 22 | 322 |
| MLP | 132 | 22 |

Table 1: training time comparison between models. Middle column states the number of epochs needed for training using early stopping. Right column states the total training time in minutes.

**(R1) "...the method may not be best suited computationally when there is high sparsity ... an incident structure is a more compact representation..."** Great Suggestion! proposing a set-to-incidence architecture is an interesting direction, however it is a slightly different problem in nature. We can list it as a future work direction.

**(R1) "...add a figure for section 4.2 which could explain the setup towards proving theorem 3."** We will make an honest effort to add an illustration explaining the structure of equation 6 which could be seen as the basis of the subsequent proofs.

**(R2) "I don't see how the set2graph formulation is clearly justified."** Indeed learning to cluster can be formulated in several ways and not necessarily as a set2graph function. Having said that, our focus is on **universal** set-2-graph functions. As far as we are aware, our method is the first one that possess this universality property and therefore an approximate arbitrary continuous clustering functions.

|  | Function | Computational | Memory |
|---|---|---|---|
| Dense | $\phi$ | $O(n \cdot d^2)$ | $O(n \cdot d)$ |
|  | $\beta$ | $O(n^2 \cdot d)$ | $O(n^2 \cdot d)$ |
|  | $\psi$ | $O(n^2 \cdot d^2)$ | $O(n^2 \cdot d)$ |
| Sparse | $\phi$ | $O(n \cdot d^2)$ | $O(n \cdot d)$ |
|  | $\beta$ | $O(e \cdot d)$ | $O(e \cdot d)$ |
|  | $\psi$ | $O(e \cdot d^2)$ | $O(e \cdot d)$ |

Table 2: Complexity analysis. For the sparse case, we assume to have $e$ edges.

**(R2) Why not use link prediction models?** Link prediction models mostly use GNNs to predict per-vertex features followed by some predictor acting on pairs of features (see e.g., Link Property Prediciton in Open Graph Benchmark by Hu et al. 2020). Our GNN baseline is therefore a standard link prediction model, where since GNN takes a graph as input we used $k$-nn graphs with $k \in \{0, 5, 10\}$.

**(R3) Is $\phi$ (from DeepSets) guaranteed to be set-to-set equivalent?** We assume the reviewer meant "equivariant". If so, yes, $\phi$ (DeepSets) is guaranteed to be set-to-set equivariant.

**(R3) How should we set the loss function?** This is application dependent: for $k = 2$, if we want to learn symmetric edge function then we back-propagate from $\frac{n \cdot (n-1)}{2}$ edge losses, and if directed edge function then $n \cdot (n - 1)$.

**(R4) Compared with S2G, S2G+ does not show obvious improvements. Could authors provide more analysis on this phenomenon?** The theoretical part of the paper implies that both S2G+ and S2G have universal approximation power, hence equivalent in that aspect. The empirical results show no obvious improvement in practice as well. We decided to include S2G+ in this experiment in order to be sure that there is no real gain (i.e., better generalization) from using the full equivariant function basis of S2G+, that grows exponentially with $k$. We will make it clearer in the final version.

**(R4) I suggest to introduce some learnable operations in $\beta$, which may be beneficial for the scalability of the method.** A very interesting future work idea, however outside the scope of our paper.

**(R4) Can the authors discuss the permutation-equivariance of the proposed method?** Our method is permutation equivariant by construction. This is one of the key design considerations, and we will make it clear in the text.

**(R4) Visualization results for the generated graphs.** We will add examples of generated graphs to the supplementary.

[Meta-Review · NeurIPS 2020]

The paper attempts make progress in the problem of learning mappings from sets of vectors to (hyper) graphs. In this regard, authors propose to learn a sequence of functions mapping a set to a tuple of k-edges. The reviewers find this formulation to be interesting. This formulation is shown to be universal as well as claimed to be parameter efficient and practical. The reviewers find the universality result to be novel, meaningful, and presented nicely, but only involved fairly standard techniques. Overall, the reviewers are happy with current version of the draft. Thus, I would be happy to recommend acceptance to NeurIPS. For the final version of the paper, please incorporate all reviewer suggested changes with the extra one page (e.g. figure size, timing results etc.). Also maybe add an remark that psi network has to be sufficiently powerful depending on the problem, which will guide other people when applying set2graph.